# A cross-cultural investigation of the short version of the Celebrity Attitude Scale (CAS-7) across five countries

Rita Horváth[1,2], David C. Watson[3], Lynn McCutcheon[4], Yohanes Budiarto[5], Róbert Urbán[2], Zsolt Demetrovics[2,6,7], Fransisca Iriani Roesmala Dewi[5], Reza Shabahang[6,8], Benyamin Mokhtari Chirani[9], Joshua L. Williams[10], Carlota Cruces Serrano[11], Nancy G. McCarley[11], Jonathan E. Roberts[11], Mara S. Aruguete[12], James D. Griffith[13], Jeanne Edman[14], Thomas Green[15], Ho Phi Huynh[16], Blaine L. Browne[17], Bethany Jurs[18], Emilia Flint[19], Michael J. Bernstein[20], Hyeyeon Hwang[21], Marc Eric S. Reyes[22], Ágnes Zsila[2,23]*

1 Doctoral School of Psychology, ELTE Eötvös Loránd University, Budapest, Hungary, 2 Institute of Psychology, ELTE Eötvös Loránd University, Budapest, Hungary, 3 MacEwan University, Edmonton, Canada, 4 North American Journal of Psychology, Winter Garden, Florida, United States of America, 5 Universitas Tarumanagara, Jakarta, Indonesia, 6 Flinders University Institute for Mental Health and Wellbeing, College of Education, Psychology and Social Work, Flinders University, Bedford Park, South Australia, Australia, 7 Centre of Excellence in Responsible Gaming, University of Gibraltar, Gibraltar, Gibraltar, 8 Department of Software Systems & Cybersecurity, Faculty of Information Technology, Monash University, Melbourne, Australia, 9 University of Guilan, Rasht, Iran, 10 Department of Criminal Justice and Criminology, Georgia Southern University, Statesboro, Georgia, United States of America, 11 Georgia Southern University, Savannah, Georgia, United States of America, 12 Lincoln University, Jefferson, Missouri, United States of America, 13 Shippensburg University, Shippensburg, Pennsylvania, United States of America, 14 Consumnes River College, Sacramento, California, United States of America, 15 Elon University, Elon, North Carolina, United States of America, 16 School of Medicine and Psychology, Australian National University, Canberra, Australia, 17 Valdosta State University, Valdosta, Georgia, United States of America, 18 Neuroscience Program, Transylvania University, Lexington, Kentucky, United States of America, 19 Black Hills State University, Spearfish, South Dakota, United States of America, 20 Psychological and Social Sciences Program, Pennsylvania State University Abington, Abington, Pennsylvania, United States of America, 21 University of Central Missouri, Warrensburg, Missouri, United States of America, 22 Department of Psychology, College of Science, University of Santo Tomas, Manila, Philippines, 23 Institute of Psychology, Pázmány Péter Catholic University, Budapest, Hungary

* zsila.agnes@ppk.elte.hu

## Abstract

### Background

Celebrity worship, conceptualized as an obsessive admiration of celebrities, has generated considerable research interest over the past two decades. Admiration towards a favorite celebrity has been commonly assessed by the 23-item Celebrity Attitude Scale (CAS). Recently, a 7-item short version (CAS-7) was developed on a representative sample of Hungarian adults. This study aimed to provide further evidence for the validity and reliability of the CAS-7 measure by extending the investigation of its factor structure to other cultures and populations.

**Data availability statement:** The datasets analyzed during the current study are available from the corresponding author upon reasonable request. Participants were informed that their data will not be made public to third party, according to the fundaments of the principal investigator's university's ethical approval (Hungary). Contact information for a non-author, institutional point of contact is Dr. Pál Kővágó (kovago.pal.lajos@btk.ppke.hu), chair of the Ethics Committee of Insttute of Psychoogy of PPCU, Faculty of Humanities and Social Sciences.

**Funding:** Reza Shabahang is supported by the Australian Government Research Training Program Scholarship (AGRTPS). This funding source has no role in covering the publication fee as it is used for an individual training program which does not cover APCs.

**Competing interests:** The authors have declared that no competing interests exist.

## Methods

Data from 4,353 participants (64.4% women, $M_{age}$ = 28.22 years, $SD$ = 11.80, age range: 14–93 years) across five countries (Canada, Hungary, Indonesia, Iran, US) were used, which was collected through online questionnaires.

## Results

Consistent with previous findings, the bifactor structure with celebrity worship as a general factor and entertainment–social and intense–pathological specific factors showed the best fit in all samples. Reliability indices for the celebrity worship general factor were good.

## Conclusions

The present findings confirmed the reliability and the consistency of the factor structure of the CAS-7 across different samples, providing further evidence for the applicability of the CAS-7 in different cultures.

## 1. Introduction

Celebrity worship, which is an excessive admiration towards celebrities [1], has been associated with poorer mental health over decades of research [2,3]. The theoretical model underlying celebrity worship proposes a hierarchical structure of the construct to explain different depths of interest in celebrities ranging from mild (e.g., loving to talk to others about a favorite celebrity) to problematic levels (e.g., feeling compelled to learn the personal habits of an admired celebrity) [1,4].

Although some assessment instruments have been developed to measure the depth of emotional involvement with celebrities (e.g., the Public Figure Preoccupation Inventory [5], the Idol Worship Questionnaire [6], the Expression of Idolization Scale [7], or the Celebrity Appeal Questionnaire [8]), these measures have rarely or not been used in further studies. Therefore, attraction to celebrities is generally assessed by the Celebrity Attitude Scale (CAS), which has been applied in more than 100 studies to date [9]. The CAS has three dimensions, reflecting the hierarchical model derived from theory. The first dimension is "Entertainment-Social" (ES), which refers to healthy fan activities such as enjoying the company of others with a similar interest of celebrities. The second dimension is "Intense-Personal" (IP) which refers to an excessive emotional involvement with the admired celebrity (e.g., if something good happens to the celebrity, the person feels as if it happened to him/her), and the third dimension, the "Borderline-Pathological" (BP) involves problematic and compulsive behaviors such as imitation [1]. The most commonly used version to assess celebrity worship is the 23-item CAS [10], but there are also 22-item and 34-item versions, which have been rarely used [4,11,12].

Common problems concerning the CAS were its length, the lack of a cut-off score to differentiate between healthy and problematic celebrity admiration [13], the poorer

internal consistency of the BP factor (Cronbach's alphas were around 0.6 or below across many studies, see [10]), and the high correlation between the IP and BP factors [14], which have been regarded as the two problematic factors, while ES theoretically represents a healthy dimension of celebrity admiration. Indeed, a brief measurement could be advantageous in studies using multiple questionnaires, so the shorter version can be administered in a shorter time. Also, motivation reduction or dropout can be avoided with brief measures in which items are less repetitive. Brief versions of self-report questionnaires can also facilitate the reduction of costs of the administration [15].

To address these concerns regarding the measurement of celebrity admiration, Zsila et al. [14] recently developed and validated a 7-item version of the CAS (CAS-7), which comprises two factors: the Entertainment-Social factor (3 items) and the "Intense-Pathological" (IPBP) factor, which combines the IP and BP factors (4 items). Regarding the structure of the CAS-7, Zsila et al. [14] found that the 2-factor and bifactor models yielded excellent fit with high internal consistency and construct validity, and the bifactor model showed a better fit to the data. The 3-factor model also had good fit indices; however, the correlation between the IP and BP factors were extremely high. Therefore, based on the findings by Zsila et al. [14], the CAS-7 is mainly unidimensional. This means that similar to the 23-item version of the CAS, the calculation of a total score and scores for the specific subscales are both meaningful when applying the CAS-7.

Zsila et al.'s [14] research on the CAS-7 was conducted on a Hungarian representative sample, while most previous studies used student or adult convenience samples. Therefore, the suitability of the CAS-7 on convenience samples, which are commonly used in celebrity worship research, needs further empirical evidence. Moreover, the CAS-7 needs further validation in cross-cultural setting to provide evidence that the CAS-7 can be used in a broader cultural context. Cross-cultural studies using the CAS have been scarce. McCutcheon et al. [16] found substantial differences across students from different universities (i.e., U.S., India) in celebrity worship. McCutcheon et al. [17] also found significant difference between African-American, Hispanic and Asian American, and White participants in celebrity worship levels. Specifically, African-American participants scored higher in the IP and BP dimensions than Hispanic and Asian American participants, while White participants had higher scores compared to African-American, Hispanic, and Asian American participants. A strong preference was also demonstrated for celebrities of the same ethnicity to their own across these groups. McCutcheon et al. [17] explained that relating to a celebrity can offer a psychological escape from the difficult reality of belonging to a minority group in a society that cannot ascertain full acceptance toward these individuals. To date, no studies have provided evidence for the appropriateness of the CAS for cross-cultural comparison purposes.

Building on previous findings, and the lack of a cross-cultural investigation of the CAS, the present study focuses specifically on the extension of the usability of the CAS across diverse populations.

The aim of the current study was to test the psychometric properties of the CAS-7 in terms of factor structure and reliability in other cultures, thereby extending its potential use. To examine the consistency of the factor structure of the CAS-7, seven samples (student and fan convenience samples) from five countries (Canada, Hungary, Indonesia, Iran, and the United States) are used. This study also investigates the usability of the CAS-7 for cross-cultural research purposes if consistency across the factor structures is confirmed. Therefore, this study retests the factor structure of the CAS-7, investigating four theoretical models (one-, two-, three-, and bifactor models based on Zsila et al. [14]) and reliability indices across seven independent samples from different cultural backgrounds. Evidence for the usability of the CAS-7 in a cross-cultural setting could contribute to a more accessible assessment of celebrity worship in future research and practice in a broader, international context.

## 2. Methods

### 2.1. Participants and procedure

The final dataset comprised 4353 participants (64.4% women, $M_{age}$ = 28.22 years, $SD$ = 11.80, age range: 14–93 years) in total from seven samples (i.e., student dataset, from general population and from several celebrity fans) from five

countries (i.e., Canada, Hungary, Indonesia, Iran, US). The five countries (i.e., Canada, Hungary, Indonesia, Iran, and the US) were selected based on data availability and existing research collaborations among research teams with significant expertise in celebrity worship research. Therefore, the selection was convenience-based. Participants were recruited using convenience sample method, they completed an online questionnaire. Data collections were between 26/11/2018 and 02/03/2024 (the latest expiration date of the most recent ethical approval: 02/02/2025, but the data collection terminated earlier), with different researchers focusing on several psychological correlates – e.g., body image and life satisfaction – and celebrity worship. Sociodemographic characteristics of the samples are presented in Table 1. A chi-square test revealed a significant difference in gender distribution across samples, $\chi^2$(12, n = 4353) = 1035.04, $p < 0.001$, with the Iranian sample having the highest proportion of male participants (72.9%) and the Indonesian sample having the highest proportion of females (85.0%). Similarly, a one-way ANOVA showed significant differences in participants' age across groups, $F$(6, 4346) = 245.74, $p < 0.001$. The youngest mean age was demonstrated in the Indonesian student sample ($M_{age}$ = 19.45, $SD$ = 1.89), while the highest mean age was found in the Hungarian fan sample ($M_{age}$ = 34.70, $SD$ = 12.40) (see SM 1). The present study was conducted in accordance with the Declaration of Helsinki and the American Psychological Association (APA). Ethical approvals were granted by the principal investigators' institutions. Specifically, Texas A&M University, San Antonio (protocol number: 2020−04); the Universitas Tarumanagara Human Research Ethics Committee (UTHREC) (protocol number: 005-UTHREC/UNTAR/II/2024); the Georgia Southern Institutional Review Board (protocol number: H21452); the Research Ethics Committee of the Institute of Psychology of Pázmány Péter Catholic University, Faculty of Humanities and Social Sciences (protocol numbers: 2021_43_m, 2023_69); the Research Ethics Board of MacEwan University (protocol number: #102087) and the IRB of Lincoln University (exempt). Written informed consent was obtained for each data collection. There were no deviations from the study protocol after approvals were obtained. All participants were informed about the study's aim and provided informed consent. Informed consent was translated from English to the local language by two independent translators of the local research teams. For underage respondents (under 18 years of age), parental approval was also acquired. The data analyzed in this study were collected between 26/11/2018 and 02/03/2024. No information was available that could personally identify participants. Data for the secondary analysis were available between 01/01/2024 and 12/04/2024.

**Table 1. Samples of the 7-item Version of the CAS (CAS-7).**

| Sample # | Country | n | Year of data collection | Type of Sample (source) | Age M; SD (Range) | Gender Female (%) |
|---|---|---|---|---|---|---|
| 1 | Canada | 252 | 2022-2023 | Students | 20.55; 3.931 (18 - 46) | 154 (61.1)[a] |
| 2 | Hungary | 295 | 2023 | Students | 22.15; 3.79 (18 - 49) | 212 (71.9) |
| 3 | Hungary | 1361 | 2022-2023 | Fan Groups (e.g., Azahriah & Desh fanok, Taylor Swift Fans Hungary) | 34.70; 14.34 (14 - 93) | 1091 (80.2) |
| 4 | Indonesia | 321 | 2023 | Students | 19.45; 1.25 (18 - 25) | 273 (85) |
| 5 | Iran | 627 | 2023 | Students | 26.65; 5.65 (18 - 35) | 166 (26.5)[b] |
| 6 | US | 570 | 2021-2023 | Students | 20.50; 4.081 (18 - 54) | 408 (71.8)[c] |
| 7 | US | 927 | 2017-2018 | General (Amazon's Mechanical Turk) | 31.59; 12.07 (18 - 78) | 501 (54.3)[d] |

*Note. N* = 4,353; Gender:

[a]Other = 3 (1.2%).

[b]Other = 4 (0.6%).

[c]Other = 2 (0.4%).

[d]Other = 5 (0.5%).

## 2.2. Measures

The 7-item version of the Celebrity Attitude Scale (CAS), which originates from the 23-item version, was used [1,4,14]. This measure contains 3 ES items and 4 IPBP items, that were rated on a 5-point Likert-scale (1 = Strongly disagree, 5 = Strongly agree). Apart from the CAS-7 measure, age and gender (1 = men, 2 = women; 3 = other) were administered in each sample. For all versions that were not originally administered in English, the questionnaires were translated and back-translated to English ensure conceptual and linguistic equivalence. Specifically, Hungarian translation was available for Hungarian participants, and Persian version was administered for Iranian participants. The items are presented in SM 1.

## 2.3. Statistical analysis

Descriptive statistics and correlations were calculated using SPSS 21.0, while structural equation modeling (SEM) was performed using Mplus 8 [18]. Confirmatory factor analysis (CFA) was conducted using a robust maximum likelihood estimator (MLR). As each country sample included more than 200 participants, group sizes exceeded established recommendations for factor analysis and SEM, ensuring sufficient statistical power and stable parameter estimates [19,20]. Based on the original paper on the CAS-7 [14], multiple factor structures were tested on the CAS-7 items. Specifically, 1-, 2-, 3-, and bifactor models were investigated using CFA – the 2-factor model contained ES and IPBP subscales, the 3-factor model was constructed from ES, IP, and BP subscales, and the bifactor model comprised specific factors ES and IPBP and a general factor of celebrity worship. The Satorra-Bentler chi-square difference-test was used to compare model fit indices [21]. The following fit indices were considered for the CFA ( [22,23]: comparative fit index (CFI; ≥ 0.95 for good, ≥ 0.90 for acceptable), Tucker-Lewis index (TLI; ≥ 0.95 for good, ≥ 0.90 for acceptable),the root-mean-square error of approximation (RMSEA; ≤ 0.06 for good, ≤ 0.08 for acceptable) with its 90% confidence interval (CI), and the standardized root-mean-square residuals (SRMR; ≤ 0.05 for good, ≤ 0.10 for acceptable). General reliability indices of the multidimensional constructs (Explained common variance (ECV), Omega, Omega Hierarchical, H) were also examined (see SM 1). To examine the cross-cultural comparability of the CAS-7, stepwise measurement invariance testing was employed using multigroup CFA [24] across six country samples in which the two-factor model showed acceptable fit. The Iranian sample was excluded from this analysis as the two-factor model did not show good fit in this sample. The grouping variable was the sample. Although the bifactor model showed superior fit compared to the two-factor model across some of the samples, due to the lack of differences in the measurement model (i.e., different item-level restrictions) which were present in the bifactor models, alongside with theoretical interpretability (see [14]), stability and replicability across the present samples, and good psychometric properties of the two-factor model, this model was selected for the invariance testing. The two-factor model was comparable across the samples as the measurement model was equal. Configural, metric, and scalar invariance were sequentially tested to detect potential measurement biases. Nonsignificant change in the commonly used fit indices indicates measurement invariance in the indicated levels [25,26]: ΔCFI ≤ 0.010; ΔTLI ≤ 0.010; and ΔRMSEA ≤ 0.015; ΔSRMR ≤ 0.030 for metric and 0.015 for scalar invariance.

## 3. Results

According to the model fit indices (see Table 2), the one-factor models representing celebrity worship as a latent construct did not fit the data in either sample. The 2-factor models with two intercorrelated celebrity worship factors (i.e., ES [Entertainment-Social] and IPBP [Intense-Pathological] – comprised items from the IP [Intense-Personal] and BP [Borderline-Pathological] theoretical factors) showed an excellent fit with the exception of the Iranian sample (n = 627). The correlations between ES and IPBP latent factors were high (between 0.70 (for Indonesia n = 321) – 0.83 (for Iran n = 627); mean $r = 0.77$). Item loadings were consistently above 0.4 in their respective factors across all samples (see SM 1). The 3-factor models also showed good fit to the data with high reliability indices of the specific factors, but with high correlations between IP and BP latent factors in all samples (between 0.76–0.98; mean $r = 0.87$). This means there is a substantial overlap between the two constructs. The bifactor model (with ES and IPBP specific factors, and celebrity worship as a general

factor) yielded excellent fit in all samples (see Table 2). Multidimensional reliability indices indicate that the contribution of specific factors, especially the IPBP, to the total variance is modest in comparison with the general factor's contribution; the ES-specific factor's contribution was higher than the IPBP's with the exception of the Iranian sample (n = 627), and the two US samples (n = 570; n = 927). In the Canadian sample (n = 252) and US student sample (n = 570), there was an overfit in the bifactor model, as this tendency is often observed in samples with a lower number of items and in bifactor models.

The internal consistency on all samples were acceptable for the ES ($\alpha = 0.71$–0.86), IPBP ($\alpha = 0.60$–0.87), global celebrity worship ($\alpha = 0.72$–0.89) dimensions across the bifactor models. In the three-factor models, the reliability of the IP factor

**Table 2. Model fit indices.**

| | models | Chi square | df | CFI | TLI | RMSEA (90% CI) | SRMR |
|---|---|---|---|---|---|---|---|
| Sample 1: Canadian student n = 252 | 1 factor | 60.967* | 14 | 0.896 | 0.844 | 0.115 (0.087-0.146) | 0.052 |
| | 2 factor | 25.657* | 13 | 0.972 | 0.955 | 0.062 (0.025-0.097) | 0.038 |
| | 3 factor | 25.954* | 11 | 0.967 | 0.937 | 0.073 (0.037-0.110) | 0.038 |
| | bifactor[a] | 5.293 | 8 | 1.000 | 1.016 | 0.000 (0.000-0.055) | 0.012 |
| Sample 2: Hungarian student n = 295 | 1 factor | 58.990* | 14 | 0.897 | 0.845 | 0.104 (0.078-0.133) | 0.053 |
| | 2 factor | 28.842* | 13 | 0.964 | 0.941 | 0.064 (0.032-0.096) | 0.033 |
| | 3 factor | 20.906* | 11 | 0.977 | 0.957 | 0.055 (0.015-0.091) | 0.027 |
| | bifactor[b] | 20.858* | 8 | 0.970 | 0.922 | 0.074 (0.036-0.113) | 0.027 |
| Sample 3: Hungarian fans n = 1361 | 1 factor | 166.406* | 14 | 0.943 | 0.914 | 0.089 (0.078-0.102) | 0.039 |
| | 2 factor | 47.213* | 13 | 0.987 | 0.979 | 0.044 (0.031-0.058) | 0.020 |
| | 3 factor | 28.215* | 11 | 0.994 | 0.988 | 0.034 (0.019-0.050) | 0.015 |
| | bifactor | 22.530* | 7 | 0.994 | 0.983 | 0.040 (0.022-0.060) | 0.013 |
| Sample 4: Indonesian student n = 321 | 1 factor | 50.383* | 14 | 0.894 | 0.841 | 0.090 (0.064-0.117) | 0.053 |
| | 2 factor | 19.760 | 13 | 0.980 | 0.968 | 0.040 (0.000-0.074) | 0.031 |
| | 3 factor | 15.957 | 11 | 0.986 | 0.972 | 0.037 (0.000-0.075) | 0.027 |
| | bifactor[c] | 10.861 | 8 | 0.992 | 0.978 | 0.033 (0.000-0.078) | 0.023 |
| Sample 5: Iranian general n = 627 | 1 factor | 268.909* | 14 | 0.810 | 0.714 | 0.170 (0.153-0.189) | 0.097 |
| | 2 factor | 178.544* | 13 | 0.876 | 0.800 | 0.143 (0.124-0.161) | 0.073 |
| | 3 factor | 50.053* | 11 | 0.971 | 0.944 | 0.075 (0.055-0.097) | 0.039 |
| | bifactor | 7.235 | 7 | 1.000 | 0.999 | 0.007 (0.000-0.050) | 0.009 |
| Sample 6: US student n = 570 | 1 factor | 94.523* | 14 | 0.918 | 0.876 | 0.100 (0.082-0.120) | 0.052 |
| | 2 factor | 26.584* | 13 | 0.986 | 0.978 | 0.043 (0.019-0.066) | 0.027 |
| | 3 factor | 23.169* | 11 | 0.988 | 0.980 | 0.044 (0.018-0.069) | 0.025 |
| | bifactor[d] | 3.997 | 8 | 1.000 | 1.011 | 0.000 (0.000-0.027) | 0.009 |
| Sample 7: US general n = 927 | 1 factor | 298.116* | 14 | 0.887 | 0.830 | 0.148 (0.134-0.163) | 0.061 |
| | 2 factor | 70.855* | 13 | 0.977 | 0.963 | 0.069 (0.054-0.086) | 0.029 |
| | 3 factor | 57.194* | 11 | 0.982 | 0.965 | 0.067 (0.051-0.085) | 0.026 |
| | bifactor[e] | 28.270* | 8 | 0.992 | 0.979 | 0.052 (0.032-0.074) | 0.013 |

*Note.*

* = $p < 0.05$;

[a] = residual covariance of ES1 was fixed to 0;

[b] = residual covariance of BP1 was fixed to 0;

[c] = residual covariance of IP1 was fixed to 0;

[d] = residual covariance of ES2 was fixed to 0;

[e] = residual covariance of ES2 was fixed to 0. CFI = Comparative Fit Index; TLI = Tucker-Lewis Index; RMSEA = Root Mean Square Error of Approximation; SRMR = Standardized Root Mean Square Residual.

was mostly acceptable across the samples, except for the Indonesian sample (α = 0.57) and the Iranian sample (α = 0.58). For the BP separate factor, it was only acceptable for the Iranian sample (α = 0.75), the Hungarian fans sample (α = 0.60) and the US general sample (α = 0.76; see Table 3).

In the final step, the 2-factor models with the bifactor models were compared using the Satorra-Bentler chi-square difference-test, as the 3-factor models could not be further considered due to the high inter-factor correlations between IP and BP. The only exception was for the Iranian sample (n = 627), as only the 3-factor model fitted to the data. At the same time, the inter-factor correlations between IP and BP were also high here, so only the bifactor model was appropriate for the Iranian sample (n = 627). The bifactor model showed a better fit on the Canadian sample (n = 252), Hungarian general sample (n = 1361), US student sample (n = 570), and US general sample (n = 927), but was not showing a significantly better fit on the Hungarian student sample (n = 295) and the Indonesian sample (n = 321).

Measurement invariance testing based on the two-factor model across the six country samples did not support either metric or scalar invariance (see Table 4). Specifically, changes in the fit indices (i.e., CFI, TLI, RMSEA, SRMR) were substantial across the scalar–metric, and metric–configural models. Therefore, measurement invariance was not demonstrated across the samples. In more detail, the factor loadings and the intercepts (mean scores) are substantially varied across the samples, preventing meaningful cross-cultural comparison in celebrity worship levels measured by the CAS-7. In summary, the CAS-7 measure does not meet the criteria for cross-cultural comparability in terms of mean scores and strength of associations between celebrity worship and other constructs, despite the consistent and replicable factor structures across the cultural samples.

**Table 3. Descriptive Statistics of the CAS-7 across the samples.**

|  | Sample 1: Canadian student n = 252 | Sample 2: Hungarian student n = 295 | Sample 3: Hungarian fans n = 1361 | Sample 4: Indonesian student n = 321 | Sample 5: Iranian general n = 627 | Sample 6: US student n = 570 | Sample 7: US general n = 927 |
|---|---|---|---|---|---|---|---|
| **Mean** | 17.11 | 15.80 | 18.83 | 23.21 | 14.74 | 16.39 | 19.41 |
| **SD** | 5.28 | 5.42 | 6.47 | 4.26 | 5.66 | 5.56 | 7.09 |
| **Range** | 7-35 | 7-35 | 7-35 | 7-33 | 7-35 | 7-33 | 7-35 |
| **Cronbach α** | | | | | | | |
| ES | 0.76 | 0.71 | 0.74 | 0.72 | 0.86 | 0.80 | 0.82 |
| IP | 0.63 | 0.63 | 0.76 | 0.57 | 0.58 | 0.64 | 0.82 |
| BP | 0.44 | 0.59 | 0.60 | 0.39 | 0.75 | 0.47 | 0.76 |
| IPBP | 0.72 | 0.72 | 0.78 | 0.60 | 0.71 | 0.72 | 0.87 |
| CAS-7 | 0.82 | 0.79 | 0.84 | 0.73 | 0.84 | 0.72 | 0.89 |

Note. ES = Entertainment-Social; IP = Intense-Personal; BP = Borderline-Pathological; IPBP = Intense-Pathological; CAS-7 = global factor

**Table 4. Testing measurement invariance on the Celebrity Attitude Scale-7.**

| Model | χ² (df) | CFI | TLI | RMSEA [90% CI] | SRMR | Model comparison | Δχ² (Δdf) | ΔCFI | ΔTLI | ΔRMSEA | ΔSRMR |
|---|---|---|---|---|---|---|---|---|---|---|---|
| Configural | 224.258 (78)*** | 0.981 | 0.969 | 0.055 [0.047-0.063] | 0.027 | | | | | | |
| Metric | 381.053 (103)*** | 0.963 | 0.955 | 0.066 [0.059-0.073] | 0.060 | metric vs. configural | 156.795 (25) | 0.018 | 0.014 | 0.011 | 0.033 |
| Scalar | 1218.239 (128)*** | 0.987 | 0.986 | 0.117 [0.111-0.123] | 0.076 | scalar vs. metric | 837.186 (25) | 0.024 | 0.031 | 0.051 | 0.016 |

Notes. ***$p < 0.001$ **$p < 0.01$; *$p < 0.05$

## 4. Discussion

Celebrity worship has been a subject of extensive research in the fields of clinical, health, and personality psychology [10]. Admiration towards celebrities is generally assessed using the 23-item Celebrity Attitude Scale (CAS; [1]). Recently, a short 7-item version of the CAS has been developed, demonstrating robust psychometric properties in terms of factor structure, reliability, and concurrent validity in a sample of Hungarian adults [14]. However, to ensure the broader appropriateness and generalizability of previous findings regarding this brief measure, further validation across diverse cultural contexts is necessary. Therefore, further evidence is needed that the CAS-7 is a reliable and valid measure with consistent factor structure across different cultural contexts. Such evidence could provide further support for the international utility of the CAS-7 in different cultures.

In the present study, the consistency of the factor structure and reliability of the CAS-7 were tested across seven convenience samples from five countries. Additionally, it was explored whether the CAS-7 could be appropriate for cross-cultural comparisons. According to the findings, the bifactor model showed a good fit, consistently across all samples. Consistent with the original structure of the CAS-7 [14], this model comprised a 'celebrity worship' general factor with two specific factors: Entertainment-Social (ES) and Intense-Pathological (IPBP). Reliability indices of the general and specific factors were satisfactory across all samples. Therefore, the present findings also indicated that the CAS-7 is primarily a unidimensional construct, consistent with Zsila et al. [14]. Researchers should use the subscale or the total score of the CAS-7 based on the study's objectives. Specifically, subscales could be used to provide a more differentiated picture on the associations of specific behavioral or emotional dimensions (e.g., excessive absorption or healthy interest in a celebrity) with other psychological constructs (e.g., cognitive functioning), and the global score for assessing celebrity admiration when the research aim is rather focused on measuring the extent of celebrity admiration generally (see [14]). While its applicability for cross-cultural comparisons was not supported in the present study, the CAS-7 may nonetheless prove to be a valuable tool in psychological practice and research for identifying individuals with elevated celebrity worship tendencies and exploring their mental health correlates. Besides its serious limitations regarding cultural non-invariance, the scale's use in different cultural contexts also requires careful interpretation, as cultural norms may shape both fandom expressions and thresholds for considering a fan engagement as problematic.

Also, the two-factor model showed a good fit in all cases, except for the Iranian sample. In the study by Zsila et al. [14], the two-factor model also showed a good fit; however, the bifactor model had a closer fit, which was also demonstrated in four samples in this study (i.e., Canadian, Hungarian general, US student, US general samples). However, the superiority of the bifactor model over the two-factor model was not demonstrated in the Hungarian students and the Indonesian samples. Similar to the study by Zsila et al. [14], inter-factor correlations between ES and IPBP were high. The two-factor model did not show good fit to the data in the Iranian sample, which may be attributed to methodological issues (e.g., small sample size) rather than fundamental cultural differences, as evidenced by the bifactor model's robust fit. The 1-factor models representing celebrity worship as a latent construct did not fit the data in either sample, which is consistent with the results by Zsila et al. [14].

In summary, the present findings suggest that the CAS-7 is a psychometrically sound tool for assessing celebrity worship separately in diverse cultural contexts, as the bifactor structure demonstrated good overall fit across each sample. Measurement invariance testing did not support metric and scalar invariance, suggesting that neither factor loadings, nor intercepts were equivalent across the samples, seriously limiting the scale's appropriateness for cross-cultural comparisons in terms of associations or latent means.

In brief, the current research shows that the CAS-7 had a consistent bifactor structure with high internal consistency across all seven samples from five countries. However, the CAS-7 is not appropriate for cross-cultural comparisons. Future studies should develop a measure that allows for cross-cultural research purposes.

### 4.1. Limitations

The concept of 'celebrity' itself may vary considerably across cultures, influencing how celebrity admiration is experienced and reported [27,28]. In English-speaking areas or in the Spanish culture, there are world-famous celebrities, while in some smaller countries or subcultures, local musicians may receive a different appreciation from their fans [29]. McCutcheon et al. [1] pointed out the role of the cultural milieu in celebrity admiration, suggesting that the education largely determines who people value, or choose as their favorite celebrity, or even who they have access to. It may be an important background factor explaining why cross-cultural comparisons were not possible with the CAS-7, or may not even be meaningful, provided that the samples under investigation could be varied even within culture. Moreover, fame may be temporary, especially in the world of social media. Another factor that may prevent cross-cultural comparisons in celebrity worship is that some celebrities rapidly become famous or decrease in popularity within a country or a local social milieu, resulting in perpetual dynamic changes in the celebrity culture of a country [30].

As a limitation of the study, the convenience sampling method can be mentioned, which did not allow for drawing general conclusions regarding the whole population of the participating countries. The use of convenience sampling restricts the generalizability of the results, as the sample may not be representative of the broader populations in the participating countries. The different sample sizes could be another limitation. As cross-cultural comparisons were not possible due to the divergence across the measurement models, estimating cut-off points on the CAS-7 across samples was not meaningful as that would be applicable only on that given, convenience sample dataset, which would not be suitable for generalization. Another important limitation of this study relates to the heterogeneity of the samples. Demographic variability may further limits the generalizability of the present findings. Additional variability may have arisen from the inclusion of fan, general population, and student samples, which may differ not only in demographic characteristics but also in the level of engagement with the celebrity culture and patterns of popular media content consumption. Finally, neither metric, nor scalar invariance was supported across the examined country samples. Items and item groups contributing to the lack of measurement invariance were investigated by analyzing the invariance of the two factors of the CAS-7 separately. Evidence of metric or scalar invariance was not found for either factor. Therefore, we were unable to identify any item group that could be invariant across the examined samples. Future research may consider applying further item-level analysis (e.g., item response theory methods).

### 4.2. Conclusions

Overall, the current study provides some preliminary evidence supporting the potential applicability of the CAS-7 in diverse cultural contexts separately. Although, there is a need for an assessment instrument that can measure celebrity admiration invariantly in different cultures or even subcultures. To investigate how universal celebrity worship can be across cultures, an appropriate measure is needed. Despite this limitation, preliminary evidence from five countries suggests that the CAS-7 has a consistent bifactor structure and good reliability, confirming its suitability for research purposes and practical utilization in a wider international context. Given that the bifactor model demonstrated consistently good fit across all samples, this model structure can be used in various cultural contexts without the need for any culture-specific modifications. However, the lack of both metric and scalar invariance indicates that future studies may benefit from investigating the cross-cultural application of the CAS-7 further by analyzing its psychometric properties and testing invariance in other cultural contexts (e.g., more European and Asian countries) to broaden cultural usability perspectives.

### Supporting information

**S1 Table. SM Table 1: Items of the 7-item version of the CAS (CAS-7).** Note. *ES = Entertainment-Social; IP = Intense-Personal; BP = Borderline-Pathological.*
(DOCX)

**S2 Table.  SM Table 2: 1-factor model: Factor loadings.** Note. *ES = Entertainment-Social; IP = Intense-Personal; BP = Borderline-Pathological.*
(DOCX)

**S3 Table.  SM Table 3: 2-factor model: Factor loadings.** Note. *ES = Entertainment-Social; IP = Intense-Personal; BP = Borderline-Pathological; IPBP = Intense-Pathological.*
(DOCX)

**S4 Table.  SM Table 4: 3-factor model: Factor loadings.** Note. $r_{ES\text{-}IP}$ = correlation between Entertainment-Social and Intense-Personal factors. $r_{ES\text{-}BP}$ = correlation between Entertainment-Social and Borderline-Pathological factors. $r_{IP\text{-}BP}$ = correlation between Intense-Personal and Borderline-Pathological factors.
(DOCX)

**S5 Table.  SM Table 5: Bifactoral model: Factor loadings.** *Note. ES = Entertainment-Social; IP = Intense-Personal; BP = Borderline-Pathological.*
(DOCX)

**S6 Table.  SM Table 6: Reliability indices of the bifactor model in seven samples.** *Note. ES=Entertainment-Social;IPBP=Intense-Pathological; CAS7=general factor; ECV=Explained Common Variance: is the proportion of all common variance explained by that factor; Ω=Omega: is model-based estimate of internal reliability of the multidimensional composite; Ωh=Omega Hierarchical: reflects the percentage of systematic variance in unit-weighted total scores that can be attributed to the individual differences ont he general factors; H=represents the correlation between a factor and an optimally-weighted item composite.*
(DOCX)

**S7 Table.  SM Table 7: Chi-square test with post-hoc z-tests for gender differences across study-samples.** *Note.* Chi-square test: $\chi^2(12, n = 4353) = 1035.04$, $p < 0.001$; Identical letters in the same row indicate nonsignificant ($p > 0.05$) difference between groups, while different letters indicate significant group differences ($p < 0.05$) according to the post-hoc z-test.
(DOCX)

## Author contributions

**Conceptualization:** Zsolt Demetrovics.

**Formal analysis:** Rita Horváth, Marc Eric S. Reyes.

**Investigation:** David C. Watson, Lynn McCutcheon, Yohanes Budiarto, Fransisca Iriani Roesmala Dewi, Reza Shabahang, Benyamin Mokhtari Chirani, Joshua L. Williams, Carlota Cruces Serrano, Nancy G. McCarley, Jonathan E. Roberts, Mara S. Aruguete, James D. Griffith, Jeanne Edman, Thomas Green, Ho Phi Huynh, Blaine L. Browne, Bethany Jurs, Emilia Flint, Michael J. Bernstein, Hyeyeon Hwang.

**Methodology:** Rita Horváth, Róbert Urbán.

**Resources:** David C. Watson, Lynn McCutcheon, Yohanes Budiarto, Fransisca Iriani Roesmala Dewi, Reza Shabahang, Benyamin Mokhtari Chirani, Joshua L. Williams, Carlota Cruces Serrano, Nancy G. McCarley, Jonathan E. Roberts, Mara S. Aruguete, James D. Griffith, Jeanne Edman, Thomas Green, Ho Phi Huynh, Blaine L. Browne, Bethany Jurs, Emilia Flint, Michael J. Bernstein, Hyeyeon Hwang.

**Supervision:** Ágnes Zsila.

**Visualization:** Rita Horváth.

**Writing – original draft:** Rita Horváth, Róbert Urbán, Zsolt Demetrovics, Ágnes Zsila.

**Writing – review & editing:** David C. Watson, Lynn McCutcheon, Yohanes Budiarto, Fransisca Iriani Roesmala Dewi, Reza Shabahang, Benyamin Mokhtari Chirani, Joshua L. Williams, Carlota Cruces Serrano, Nancy G. McCarley, Jonathan E. Roberts, Mara S. Aruguete, James D. Griffith, Jeanne Edman, Thomas Green, Ho Phi Huynh, Blaine L. Browne, Bethany Jurs, Emilia Flint, Michael J. Bernstein, Hyeyeon Hwang, Marc Eric S. Reyes.

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
