## [Decision Letter · Decision Letter 0]

29 Apr 2025

PONE-D-24-58562A cross-cultural investigation of the short version of the Celebrity Attitude Scale (CAS-7) across five countriesPLOS ONE

Dear Dr. Zsila,

Thank you for submitting your manuscript to PLOS ONE. After careful consideration, we feel that it has merit but does not fully meet PLOS ONE’s publication criteria as it currently stands. Therefore, we invite you to submit a revised version of the manuscript that addresses the points raised during the review process.

We look forward to receiving your revised manuscript.

Kind regards,

Osmond Ekwebelem

Academic Editor

PLOS ONE

Journal Requirements:

 “Reza Shabahang is supported by the Australian Government Research Training Program Scholarship (AGRTPS). This funding source has no role in covering the publication fee as it is used for an individual training program which does not cover APCs.”

“NO”

5. In this instance it seems there may be acceptable restrictions in place that prevent the public sharing of your minimal data. However, in line with our goal of ensuring long-term data availability to all interested researchers, PLOS’ Data Policy states that authors cannot be the sole named individuals responsible for ensuring data access (http://journals.plos.org/plosone/s/data-availability#loc-acceptable-data-sharing-methods).

6. PLOS requires an ORCID iD for the corresponding author in Editorial Manager on papers submitted after December 6th, 2016. Please ensure that you have an ORCID iD and that it is validated in Editorial Manager. To do this, go to ‘Update my Information’ (in the upper left-hand corner of the main menu), and click on the Fetch/Validate link next to the ORCID field. This will take you to the ORCID site and allow you to create a new iD or authenticate a pre-existing iD in Editorial Manager.

8. Please include your tables as part of your main manuscript and remove the individual files. Please note that supplementary tables (should remain/ be uploaded) as separate "supporting information" files.

Additional Editor Comments:

• Some sentences are unnecessarily long and complex. Breaking them down would improve readability.

Example: "Therefore, based on the findings by Zsila et al. (14), the CAS-7 is fundamentally unidimensional with some multidimensionality." Simplify or rephrase to avoid confusion between "fundamentally unidimensional" and "some multidimensionality."

• Some minor errors, like missing commas or awkward phrasing.

Examples:

o Line 8: "Cronbach’s 123 alphas" = Should just be "Cronbach’s alphas."

o Line 9: "which have been generally used in celebrity worship research." It could be smoother as "which are commonly used in celebrity worship research."

• Terms like "cross-cultural setting" and "applicability" are repeated multiple times. Consider varying word choice slightly for flow.

• While you hint at cultural differences (e.g., McCutcheon et al. finding differences between U.S. and India), you don't explain why cultural context might impact celebrity worship. Briefly explaining this would better justify the need for cross-cultural validation.

• The transition to study aims feels a little abrupt. A bridge sentence summarizing why cross-cultural validation is needed based on previous limitations could strengthen it.

Questions for the Authors on introduction:

• How were the five countries (Canada, Hungary, Indonesia, Iran, and the U.S.) selected? Were there any cultural frameworks (e.g., individualism-collectivism) considered in their selection?

• Since you're using student and fan samples, how comparable are these samples across countries? Are they matched on key demographics (age, gender)?

• Was measurement invariance testing (e.g., configural, metric, scalar invariance) considered to formally establish cross-cultural comparability of the CAS-7?

• If CAS-7 is found to be "fundamentally unidimensional," how should future researchers treat the subscales — still interpret separately or focus mainly on the total score?

• If CAS-7 shows poor fit in certain countries, would you recommend culturally adapting it, or would you suggest developing entirely new culturally sensitive scales?

Comments on the methods

• The word “database” sounds strange when referring to people (line 168). Consider replacing it with "sample" or "dataset".

• You mention the 3 ES and 4 IPBP items but don't explain what ES and IPBP stand for here. This should be briefly expanded for clarity.

Questions for the Authors on methods:

• Convenience sampling inherently limits generalizability, but this limitation is not acknowledged. It should be.

• How many participants came from each country/sample (students vs general population vs celebrity fans)?

• Was the sample distribution approximately equal across groups, or heavily skewed toward one?

• It is unclear why data access dates are different from data collection dates.

o Line 182–183: "The data for research purposes were accessed between 31/12/2015 and 12/04/2024." But participants were recruited starting from 2018. Why access data from 2015?

o Please clarify whether there were existing datasets before new data collection started.

• Other than age and gender, were any other sociodemographic variables collected (e.g., education, income, country of residence)? If yes, mention them.

• Why is this large sample appropriate for CFA and SEM? (A brief note would strengthen the methods section.)

• Why are the data access dates (starting from 2015) earlier than the participant recruitment dates (2018)? Were earlier datasets used? Please clarify.

• Since participants were from five different countries, was the CAS-7 translated into local languages?

• Could you please define ES and IPBP when first introducing them? It will help readers unfamiliar with the original CAS structure.

Comments on the discussion

• Ideas sometimes jump between results, limitations, and broader interpretations without clear transitions. Use more structured subsections or clearer paragraph transitions.

• The claim that CAS-7 “may be widely applicable” (lines 310–311) feels overstated, especially given the cross-cultural limitations admitted. Temper this language to avoid implying broader validity than supported.

• The Iranian sample's deviation is mentioned but not discussed in depth. Elaborate more on why the Iranian sample's results differed—was it sample size, cultural interpretation of celebrity, translation issues, etc.?

• Some sentences are awkward or confusing, particularly around lines 288–293. Example: “Looking at the term ‘celebrity’, it may have a different meaning across cultures.”

Reword to, e.g., "The concept of 'celebrity' itself may vary considerably across cultures, influencing how celebrity admiration is experienced and reported."

• Besides research, could CAS-7 be used for clinical, media, or public health purposes in different countries? Briefly discuss practical applications and caution about their limitations due to cultural variability.

• It’s unclear how different participant sources (students, general public, celebrity fans) might have influenced the findings. Reflect briefly on whether sample type (beyond nationality) may have introduced additional variability.

Questions for the Authors on the discussion:

• What are your hypotheses for why the two-factor model did not fit well for the Iranian sample? Could it be related to cultural norms about fame or admiration?

• Were there particular CAS-7 items that performed inconsistently across cultures? If yes, which ones and why might that be?

Reviewers' comments:

Reviewer's Responses to Questions

**Comments to the Author**

1. Is the manuscript technically sound, and do the data support the conclusions?

Reviewer #1: Yes

2. Has the statistical analysis been performed appropriately and rigorously? 

Reviewer #1: Yes

3. Have the authors made all data underlying the findings in their manuscript fully available?

Reviewer #1: Yes

4. Is the manuscript presented in an intelligible fashion and written in standard English?

Reviewer #1: Yes

5. Review Comments to the Author

Reviewer #1: Dear authors, I found your work invaluable and a timely issue however I have a suggestion for you. Despite mentioned in problem statement slightly, you did not justify 7-point likert scale than a 5-point likert scale. I believe that you need to critically show the rationale of using CAS-7 factor structure with its implications under the method section-measure section.

6. PLOS authors have the option to publish the peer review history of their article (what does this mean? ). If published, this will include your full peer review and any attached files.

**Do you want your identity to be public for this peer review?** For information about this choice, including consent withdrawal, please see our Privacy Policy .

Reviewer #1: No

---

## [Author Response · Author response to Decision Letter 1]

20 Jul 2025

Dear Osmond Ekwebelem,

We are grateful for the positive and constructive comments. The insightful suggestions helped us significantly improve the quality of our manuscript. Following your recommendation, we are pleased to submit a revised version of the manuscript entitled “A cross-cultural investigation of the short version of the Celebrity Attitude Scale (CAS-7) across five countries” for further consideration in PlosOne.

We have addressed all the comments (summarized below), which you can see in the revised version of the manuscript with yellow highlights. Reviewer comments appear in bold, followed by our response, and the revised section of the manuscript (in italics).

Again, we are grateful for all the recommendations. We hope that the revisions will meet your standards for publication in PlosOne.

Sincerely,

the authors

Editor

Comments

1.Some sentences are unnecessarily long and complex. Breaking them down would improve readability.

Example: "Therefore, based on the findings by Zsila et al. (14), the CAS-7 is fundamentally unidimensional with some multidimensionality." Simplify or rephrase to avoid confusion between "fundamentally unidimensional" and "some multidimensionality."

Author’s response:

Thank you for bringing this to our attention. We have made the necessary modifications in the manuscript accordingly.

Changes in themanuscript:

Page 8:

“Therefore, based on the findings by Zsila et al. (14), the CAS-7 is mainly unidimensional.”

Page 15:

“Therefore, the present findings also indicated that the CAS-7 is primarily a unidimensional construct, consistent with Zsila et al. (14).”

2.Some minor errors, like missing commas or awkward phrasing.

Examples:

o Line 8: "Cronbach’s 123 alphas" = Shouldjust be "Cronbach’s alphas."

o Line 9: "which have been generally used in celebrity worship research." It could be smoother as "which are commonly used in celebrity worship research."

Author’s response:

We carefully revised and rewrite those lines accordingly.

Changes in themanuscript:

Page 8:

„Common problems concerning the CAS were its length, the lack of a cut-off score to differentiate between healthy and problematic celebrity admiration (13), the poorer internal consistency of the BP factor (Cronbach’s alphas were around 0.6 or below across many studies, see [10]), and the high correlation between the IP and BP factors (14), which have been regarded as the two problematic factors, while ES theoretically represents a healthy dimension of celebrity admiration.”

Page 9:

“Therefore, the suitability of the CAS-7 on convenience samples, which are commonly used in celebrity worship research, needs further empirical evidence.”

3.Terms like "cross-cultural setting" and "applicability" are repeated multiple times. Consider varying word choice slightly for flow.

Author’sresponse:

Thank you for pointing this out. We made some changes in the manuscrip taccordingly.

Changes in themanuscipt:

Page 8-9:

“Therefore, the suitability of the CAS-7 on convenience samples, which are commonly used in celebrity worship research, needs further empirical evidence.”

Page 9:

„To date, no studies have provided evidence for the appropriateness of the CAS for cross cultural comparison purposes.”

“The aim of the current study was to test the psychometric properties of the CAS-7 in terms of factor structure and reliability in other cultures, thereby extending its potential use. To examine the consistency of the factor structure of the CAS-7, seven samples (student and fan convenience samples) from five countries (Canada, Hungary, Indonesia, Iran, and the United States) are used. This study also investigates the usability of the CAS-7 for cross-cultural research purposes if consistency across the factor structures is confirmed.”

Page 10:

“Evidence for the usability of the CAS-7 in a cross-cultural setting could contribute to a more accessible assessment of celebrity worship in future research and practice in a broader, international context.”

Page 14:

“However, to ensure the broader appropriateness and generalizability of previous findings regarding this brief measure, further validation across diverse cultural contexts is necessary.”

“Such evidence could provide further support for the international utility of the CAS-7 in different cultures.”

Page 16:

“Despite this limitation, preliminary evidence from five countries suggests that the CAS-7 has a consistent bifactor structure and good reliability, confirming its suitability for research purposes and practical utilization in a wider international context.”

4.While you hint at cultural differences (e.g., McCutcheon et al. finding differences between U.S. and India), you don't explain why cultural context might impact celebrity worship. Briefly explaining this would better justify the need for cross-cultural validation.

Author’s response:

Thank you for this insightful observation regarding the need to elaborate on how cultural context may impact celebrity worship. We have now elaborated more on this aspect.

Changes in the manuscript:

Page 9:

“Cross-cultural studies using the CAS have been scarce. McCutcheon et al. (16) found substantial differences across students from different universities (i.e., U.S., India) in celebrity worship. McCutcheon et al. (17) also found significant difference between African-American, Hispanic and Asian American, and White participants’ in celebrity worship levels. Specifically , African-American participants scored higher in the IP and BP dimensions than Hispanic and Asian American participants, while White participants had higher scores compared to African-American, Hispanic and Asian American participants. A strong preference was also demonstrated for celebrities of the same ethnicity to their own across these groups. McCutcheon et al. (17) explained that relating to a celebrity can offer a psychological escape from the difficult reality of belonging to a minority group in a society that cannot ascertain full acceptance toward these individuals.”

5.The transition to study aims feels a little abrupt. A bridge sentence summarizing why cross-cultural validation is needed based on previous limitations could strengthen it.

Author’s response:

Thank you for pointing this out. We integrated a bridge sentence.

Changes to the Manuscript:

Page 9:

„Building on previous findings, and the lack of a cross-cultural investigation of the CAS, the present study focuses specifically on the extension of the usability of the CAS across diverse populations.”

Introduction

1. How were the five countries (Canada, Hungary, Indonesia, Iran, and the U.S.) selected? Were there any cultural frameworks (e.g., individualism-collectivism) considered in their selection?

Author’s response:

Thank you for your thoughtful question regarding the selection of countries. The five countries (Canada, Hungary, Indonesia, Iran, and the U.S.) were chosen based on the availability of relevant datasets and our established collaborations with research teams in these locations who are engaged in the study of celebrity worship and utilize the Celebrity Attitude Scale. The selection was therefore primarily based on convenience, reflecting accessible data sources and existing academic partnerships, rather than being guided by a specific cultural framework such as individualism-collectivism.

Changes to the manuscript:

Page 10:

„The five countries (i.e., Canada, Hungary, Indonesia, Iran, and the USA) were selected based on existing research collaborations among research teams with significant expertise in celebrity worship research. Therefore, the selection was convenience-based.”

2. Since you're using student and fan samples, how comparable are these samples across countries? Are they matched on key demographics (age, gender)?

Author’s response:

Thank you for raising this important point. To evaluate demographic comparability, we examined gender and age distributions across the seven samples. Chi-square and post hoc z-tests revealed significant differences in gender ratios with female proportions ranging from 27.1% (Iran) to 85.0% (Indonesia).

In terms of age, a one-way ANOVA showed significant differences across the samples in terms of age, F(6, 4346) = 245.74, p< .001. Tukey post hoc tests indicated that participants from Hungary (fan sample; M = 34.70) and the U.S. (general sample; M = 31.59) were significantly older than those in Indonesia (M = 19.45), U.S. students (M = 20.50), and Canada (M = 20.55).

These findings suggest that the samples had significant differences on key demographic variables. We now report these findings (see SM Table 7) acknowledge this in the revised manuscript and recommend that future studies either recruit more demographically balanced samples or include statistical controls for age and gender when examining cross-cultural effects.

Changes to the manuscript:

Page 10:

„A chi-square test revealed a significant difference in gender distribution across samples, χ²(12, n = 4353) = 1035.04, p < 0.001, with the Iranian sample having the highest proportion of male participants (72.9%) and the Indonesian sample having the highest proportion of females (85.0%). Similarly, a one-way ANOVA showed significant differences in participants’ age across groups, F(6, 4346) = 245.74, p < 0.001. The youngest mean age was demonstrated in the Indonesian student sample (Mage = 19.45, SD = 1.89), while the highest mean age was found in the Hungarian fan sample (Mage = 34.70, SD = 12.40) (see SM Table 7).”

Page 17:

“Another important limitation of this study relates to the heterogeneity of the samples. Demographic variability may further limits the generalizability of the present findings.”

3. Was measurement invariance testing (e.g., configural, metric, scalar invariance) considered to formally establish cross-cultural comparability of the CAS-7?

Author’s response:

Thank you for this valuable comment. To assess cross-cultural comparability of the CAS-7, we conducted measurement invariance testing using a stepwise approach (configural, metric, scalar) across six country samples where the two-factor model showed acceptable fit. While the configural model demonstrated good fit (χ²(78) = 224.26, p < 0.001; CFI = .981; RMSEA = .055), subsequent models indicated declining fit indices. The metric model showed moderate change (ΔCFI = .018; ΔRMSEA = .011), and the scalar model yielded a notable decline in fit (ΔCFI = .024; ΔRMSEA = .051), suggesting that full scalar invariance was not supported. Given these findings and the known limitations of bifactor models in cross-cultural comparisons due to differential constraints, we opted to retain the more stable and theoretically grounded two-factor structure for the current study. These limitations and theoretical implications are now discussed in the manuscript.

Changes to the manuscript:

Page 12-13:

“To examine the cross-cultural comparability of the CAS-7, stepwise measurement invariance testing was employed using multigroup CFA (24) across six country samples in which the two-factor model showed acceptable fit. The Iranian sample was excluded from this analysis as the two-factor model did not show good fit in this sample. The grouping variable was the origin of the sample (i.e., country). Although the bifactor model showed superior fit compared to the two-factor model across some of the samples, due to the lack of differences in the measurement model (i.e., different item-level restrictions) which were present in the bifactor models, alongside with theoretical interpretability (see 14), stability and replicability across the present samples, and good psychometric properties of the two-factor model, this model was selected for the invariance testing. The two-factor model was comparable across the samples as the measurement model was equal. Configural, metric, and scalar invariance were sequentially tested to detect potential measurement biases. Nonsignificant change in the commonly used fit indices indicates measurement invariance in the indicated levels (25,26): ΔCFI ≤ 0.010; ΔTLI ≤ 0.010; and ΔRMSEA ≤ 0.015; ΔSRMR ≤ 0.030 for metric and 0.015 for scalar invariance.”

Page 15:

“Measurement invariance testing based on the two-factor model across the six country samples did not support either metric or scalar invariance (see Table 4). Specifically, changes in the fit indices (i.e., CFI, TLI, RMSEA, SRMR) were substantial across the scalar–metric, and metric–configural models. Therefore, measurement invariance was not demonstrated across the samples. In more detail, the factor loadings and the intercepts (mean scores) are substantially varied across the samples, preventing meaningful cross-cultural comparison in celebrity worship levels measured by the CAS-7. In summary, the CAS-7 measure does not meet the criteria for cross-cultural comparability in terms of mean scores and strength of associations between celebrity worship and other constructs, despite the consistent and replicable factor structures across the cultural samples.”

Page 16:

“In summary, the present findings suggest that the CAS-7 is a psychometrically sound tool for assessing celebrity worship separately in diverse cultural contexts, as the bifactor structure demonstrated good overall fit across each sample. Measurement invariance testing did not support metric and scalar invariance, suggesting that neither factor loadings, nor intercepts were equivalent across the samples, seriously limiting the scale’s appropriateness for cross-cultural comparisons in terms of associations or latent means.”

Page 18:

“Finally, neither metric, nor scalar invariance was supported across the examined country samples. Items and item groups contributing to the lack of measurement invariance were investigated by analyzing the invariance of the two factors of the CAS-7 separately. Evidence of metric or scalar invariance was not found for either factor. Therefore, we were unable to identify any item group that could be invariant across the examined samples. Future research may consider applying further item-level analysis (e.g., item response theory methods).”

4. If CAS-7 is found to be "fundamentally unidimensional," how should future researchers treat the subscales — still interpret separately or focus mainly on the total score?

Author’s response:

Thank you for raising this important methodological consideration. While the CAS-7 demonstrates fundamental unidimensionality, we recommend that researchers interpret both subscales (entertainment-social and intense-pathological) alongside the total score. These subscales capture distinct dimensions of celebrity worship (e.g., social bonding vs. excessive preoccupation), which can remain meaningful despite the scale’s unidimensional structure, based on the respective research aims. The choice to prioritize subscales or the global score should depend on the study’s focus: subscales are preferable for examining mechanisms linked to specific behaviors (e.g., differentiating healthy engagement from problematic levels of admiration), while the total score is suitable for assessing general celebrity admiration. This approach aligns with recent psychometric recommendations advocating for context-driven interpretations of unidimensional measures (see Zsila et al., 2024). We have clarified this rationale in the revised manuscript.

Changes to the manuscript:

Page 15:

"Researchers should use the subscale or the total score of the CAS-7 based on the study’s objectives. Specifically, subscales could be used to provide a more differentiated picture on the associations of specific behavioral or emotional dimensions (e.g., excessive absorption or healthy interest in a celebrity) with other psychological constructs (e.g., cognitive functioning), and the global score for assessing celebrity admiration when the research aim is rather focused on measuring the extent of celebrity admiration generally (see (14)).

5. If CAS-7 shows poor fit in certain countries, would you recommend culturally adapting it, or would you suggest developing entirely new culturally sensitive scales?

Author’s response:

Thank you for raising this critical methodological question. Our analyses revealed no instances of poor model fi

---

## [Editor Report · Decision Letter 1]

20 Aug 2025

A cross-cultural investigation of the short version of the Celebrity Attitude Scale (CAS-7) across five countries

PONE-D-24-58562R1

Dear Dr. Zsila,

We’re pleased to inform you that your manuscript has been judged scientifically suitable for publication and will be formally accepted for publication once it meets all outstanding technical requirements.

Kind regards,

Osmond Ekwebelem

Academic Editor

PLOS ONE

Additional Editor Comments (optional):

Comments and questions raised were thoroughly addressed.

---

## [Editor Report · Acceptance letter]

PONE-D-24-58562R1

PLOS ONE

Dear Dr. Zsila,

I'm pleased to inform you that your manuscript has been deemed suitable for publication in PLOS ONE. Congratulations! Your manuscript is now being handed over to our production team.

Kind regards,

on behalf of

Dr. Osmond Ekwebelem

Academic Editor

PLOS ONE